# Brain Protective Effect of Resveratrol via Ameliorating Interleukin-1β-Induced MMP-9-Mediated Disruption of ZO-1 Arranged Integrity

**DOI:** 10.3390/biomedicines10061270

**Published:** 2022-05-29

**Authors:** Ming-Ming Tsai, Jiun-Liang Chen, Tsong-Hai Lee, Hsiuming Liu, Velayuthaprabhu Shanmugam, Hsi-Lung Hsieh

**Affiliations:** 1Division of Basic Medical Sciences, Department of Nursing, Research Center for Chinese Herbal Medicine, Graduate Institute of Health Industry Technology, Chang Gung University of Science and Technology, Taoyuan 33303, Taiwan; mmtsai@mail.cgust.edu.tw; 2Department of General Surgery, New Taipei Municipal Tucheng Hospital, New Taipei 236017, Taiwan; 3Division of Chinese Internal Medicine, Center for Traditional Chinese Medicine, Chang Gung Memorial Hospital, School of Traditional Chinese Medicine, College of Medicine, Chang Gung University, Taoyuan 33305, Taiwan; a12015@cgmh.org.tw; 4Stroke Center and Stroke Section, Department of Neurology, Chang Gung Memorial Hospital, College of Medicine, Chang Gung University, Taoyuan 33305, Taiwan; thlee@cgmh.org.tw; 5Department of Food Science, National Taiwan Ocean University, Keelung 202301, Taiwan; hmliu@mail.ntou.edu.tw; 6Department of Biotechnology, Bharathiar University, Coimbatore 641046, India; velayuthaprabhu@buc.edu.in; 7Department of Neurology, Chang Gung Memorial Hospital, Taoyuan 33305, Taiwan

**Keywords:** resveratrol, matrix metalloproteinase-9, reactive oxygen species, brain microvascular endothelial cells, anti-inflammation, brain protection

## Abstract

In the central nervous system (CNS), the matrix metalloproteinase-9 (MMP-9) is induced by several factors and contributes to CNS disorders, including inflammation and neurodegeneration. Thus, the upregulation of MMP-9 has been considered to be an indicator of inflammation. Interleukin-1β (IL-1β) is an important proinflammatory cytokine which can induce various inflammatory factors, such as MMP-9, in many inflammatory disorders. Several phytochemicals are believed to reduce the risk of several inflammatory disorders, including the CNS diseases. Among them, the resveratrol, a principal phenolic compound of the grape, blueberry, and mulberry peels and Cassia plants, has been shown to possess several medicinal properties, including antioxidative, anti-inflammatory, and antitumor function. Herein, we used mouse-brain microvascular endothelial cells (bMECs) to demonstrate the signaling mechanisms of IL-1β-induced MMP-9 expression via zymographic, RT-PCR, Western blot, reactive oxygen species (ROS) detection, immunofluorescence stain, and promoter reporter analyses. Then we evaluated the effects of resveratrol on IL-1β-induced MMP-9 expression in bMECs and its mechanism of action. We first demonstrated that IL-1β induced MMP-9 expression in bMECs. Subsequently, IL-1β induced MMP-9 expression via ROS-mediated c-Src-dependent transactivation of EGFR, and then activation of the ERK1/2, p38 MAPK, JNK1/2, and NF-κB signaling pathway. Finally, we determined that IL-1β-induced upregulation of MMP-9 may cause the disruption of the arranged integrity of zonula occludens-1 (ZO-1), but this could be inhibited by resveratrol. These data indicated that resveratrol may have antioxidative and brain-protective activities by reducing these related pathways of ROS-mediated MMP-9 expression and tight junction disruption in brain microvascular endothelial cells.

## 1. Introduction

In the central nervous system (CNS), cerebrovascular cells such as brain microvascular endothelial cells (bMECs) play a vital role in maintaining the brain microenvironment and functions, including cerebral blood flow, microvascular tone, and blood–brain barrier (BBB) integrity [1,2]. However, dysfunction of the vascular endothelium is an early finding in the development of several vascular and cerebral diseases and is closely related to clinical events in patients with atherosclerosis, stroke, and neurodegenerative diseases [3,4,5]. The pathological changes in the integrity and function of brain microvessels can further trigger damage to brain nerve cells in the microenvironment area [3,5]. Therefore, strategies for providing cerebrovascular protection have received increased attention in the CNS disorders.

Natural medicinal plants are an important part of traditional medicine, which, today, is considered one of the most complete systems of complementary medicine. Recently, several natural products have been included into European Pharmacopoeia. The effects of several natural herbs have been demonstrated over the past decade, and phytochemicals have been shown to have long-term health-promoting or medicinal effects. Phytochemicals are believed to reduce the risk of several major diseases, including neurodegenerative disorders [6]. The likely mechanisms of the action may be maintaining redox balance and oxidative stress prevention. To date, hundreds of natural products have been demonstrated for their antioxidant properties in vitro and in vivo. Consequently, several natural products can be considered a source of potent antioxidant compounds, and this could explain part of their therapeutic and preventive utility [7]. Among these, resveratrol is a common natural polyphenol product isolated from the grape, blueberry, mulberry, peanuts, soy, and Cassia plants [8,9], and it displays a complex spectrum of pharmacology and an array of medicinal properties. It has been used as a phytoalexin or an antioxidant [10]. Moreover, resveratrol has been demonstrated to exhibit anticancer, cardioprotective, and immunomodulatory function properties [8,9,10]. In the CNS, a previous report indicated that some natural phenolic compounds, such as resveratrol, may act as potential neuroprotective agents to treat Parkinson’s disease [10] or multiple sclerosis [11]. Although there are extensive published reports that indicate the effects of resveratrol in various disorders, including the CNS disorders, the mechanisms in the CNS neuroprotective action is not clearly understood.

Matrix metalloproteinases (MMPs) are the crucial molecules for the turnover of the extracellular matrix (ECM) and pathophysiological processes; they are a large family of zinc-dependent endopeptidases [12,13]. Among them, MMP-9 has been indicated to be involved in morphogenesis, wounding healing, and neurite outgrowth in the CNS [13,14]. Several studies have demonstrated that MMP-9 may participate in several brain injuries and induce the pathogenic process of brain diseases [14,15]. Moreover, several proinflammatory factors, including cytokines and endotoxin, have been shown to upregulate MMP-9 expression and activity in brain cells [14,16]. Moreover, MMP-9 also contributes to dysfunction of the integrity of the blood–brain barrier (BBB) and then causes neuroinflammation by various harmful factors, such as lipopolysaccharide [17]. Previously, we have demonstrated that some proinflammatory mediators, such as IL-1β, induce MMP-9 expression and MMP-9-mediated effects in brain astrocytes [18,19]. These studies indicated that MMP-9 plays a key role in brain inflammation and disorders, and this has aroused our interest to study the functions of the natural phenolic compound resveratrol on MMP-9 expression in brain microvascular endothelial cells (bMECs). Here, we used the model of upregulation of MMP-9 by a pivotal proinflammatory cytokine interleukin-1β (IL-1β), in bMECs to evaluate the effects of resveratrol on MMP-9 expression and the relative events, such as tight junction damage.

Reactive oxygen species (ROS) are generated by several enzymatic and chemical processes, or are directly inhaled, including O_2_•^−^, •OH, and H_2_O_2_. The ROS have physiological roles at a low level as signaling molecules in various cellular and developmental processes and the killing of invading microorganisms [20]. In contrast, oxidative stress has been reported to play a key role in the progression of various diseases [21,22]. Moreover, ROS have been shown to interact with DNA, lipids, proteins, and carbohydrates that lead to cellular dysfunctions and inflammatory responses [21,22]. Under pathological conditions, many proinflammatory mediators induce various inflammatory genes expression during brain injury by increasing ROS generation [16,21]. Increasing evidence attributes neurodegenerative diseases such as Alzheimer’s disease (AD) to oxidative stress (generation of free radicals) that leads to brain inflammation during CNS pathogenesis [21,22,23]. Moreover, ROS also play a signaling factor mediated microglial activation induced by several proinflammatory mediators [24]. The ROS generation associated with MMP-9 has been reported in several organ diseases [16]. Our previous studies have demonstrated that ROS are critical for induction of MMP-9 responses in rat-brain astrocytes [25]. Moreover, IL-1β-induced ROS signal and MMP-9 expression in brain astrocytes imply that IL-1β-mediated redox signals and MMP-9 induction may be vital for the development of brain injuries and inflammatory diseases [19].

Based on these backgrounds and our previous studies in the brain inflammatory responses by upregulation of MMP-9 [16], the experiments were performed to evaluate the effects and signaling pathways of resveratrol on IL-1β-induced MMP-9 expression in brain microvascular endothelial cells (bMECs). In the study, we found that the resveratrol reduced IL-1β-induced MMP-9-mediated disruption of the arranged integrity of zonula occludens-1 (ZO-1). Moreover, the IL-1β-stimulated activation of several signals (e.g., ROS, c-Src, EGFR, and MAPKs) has been inhibited by the pretreatment of resveratrol. Furthermore, the resveratrol decreased ROS-mediated activation of the c-Src/EGFR/Akt cascade, MAPKs (i.e., ERK1/2, p38, and JNK1/2), and NF-κB pathway in these cells. These results suggested that resveratrol may have neuroprotective effects via its antioxidative and anti-inflammatory action to prevent the MMP-9-mediated disruption of ZO-1 arranged integrity in the CNS.

## 2. Materials and Methods

### 2.1. Materials

DMEM (Dulbecco’s modified Eagle’s medium)/F-12 medium, FBS (fetal bovine serum), and TRIzol were from Invitrogen (Carlsbad, CA, USA). The Hybond C membrane and ECL (enhanced chemiluminescence) Western blot detection system were from GE Healthcare Biosciences (Buckinghamshire, UK). The phospho-c-Src (Tyr416) (Cat. #6943), phospho-EGFR (Tyr845) (Cat. #2231), phospho-Akt (Ser473) (Cat. #4060), phospho-ERK1/2 (Thr202/Tyr204) (Cat. #4370), phospho-p38 (Thr180/Tyr182) (Cat. #4511), phospho-JNK1/2 (Thr183/Tyr185)(Cat. #4668), and phospho-p65 NF-κB (Ser536) (Cat. #3033) antibodies were from Cell Signaling (Danver, MA, USA). Anti-ZO-1 (Cat. #sc-33725) antibody was from Santa Cruz (Dallas, TX, USA). Anti-GAPDH (glyceraldehyde-3-phosphate dehydrogenase) antibody was from GeneTex (Irvine, CA, USA). MMP2/9 inhibitor (2/9i), apocynin, PP1, AG1478, SH-5, U0126, SB202190, SP600125, Bay11-7082, and were from Enzo (Farmingdale, NY, USA). BCA (Bicinchoninic acid) protein assay reagent was from Pierce (Rockford, IL, USA). IL-1β (interleuline-1β) was from R&D Systems (Minneapolis, MN, USA). NAC (N-acetyl-cysteine), enzymes, and other chemicals were from Sigma (St. Louis, MO, USA).

### 2.2. Cell Cultures and Treatments

Mouse-brain microvascular endothelial cells (bMECs: bEnd.3) were cultured as described previously [26]. Cells were grown in DMEM/F-12 containing 10% FBS and antibiotics (100 U/mL penicillin G, 100 μg/mL streptomycin, and 250 ng/mL fungizone) at 37 °C in a humidified 5% CO_2_ atmosphere. Confluence cells were released, and the cell suspension (2 × 10^5^ cells/mL) was plated onto culture plates or dishes for the measurement of protein or RNA expression, respectively. The culture medium was changed after 24 h and then every 3 days. Experiments were performed with cells from passages 5 to 13. When using the inhibitors, cells were pretreated with the inhibitor for 1 h before being incubated with IL-1β (10 ng/mL). Treatment of bMECs with these inhibitors alone had no significant effect on cell viability determined by an XTT assay (data not shown).

### 2.3. MMP Gelatin Zymography

After treatment of IL-1β, the cultured media were collected and analyzed by gelatin zymography [18]. Gelatinolytic activity was manifested as horizontal white bands on a blue background. Because cleaved MMPs were not reliably detectable, only pro-form zymogens were quantified.

### 2.4. Total RNA Extraction and Reverse Transcription-PCR Analysis

Total RNA was extracted as described previously [26]. The cDNA obtained from 0.5 μg total RNA was used as a template for PCR amplification. Oligonucleotide primers were designed on the basis of GenBank entries for mouse MMP-9 and β-actin. The primers were *MMP-9*: 5′- GCTGACTACGATAAGGACGGCA-3′ (sense) and 5′-TAGTGGTGCAGGCAGAGTAGGA-3′ (antisense); *β-actin*: 5′-AGAGGGAAATCGTGCGTGAC-3′ (sense) and 5′-CAATAGTGATGACCTGGCGT-3′ (anti-sense). The amplification was performed in 30 cycles (at 55 °C, 30 s; 72 °C, 1 min; and 94 °C, 30 s). PCR fragments were determined on 2% agarose 1X TAE gel containing ethidium bromide by the molecular weight markers. The β-actin, an internal reference RNA, was amplified in parallel, and cDNA amounts were standardized to equivalent β-actin mRNA levels. The images were quantified and analyzed by an UN-SCAN-IT gel version 6.1 software (Silk Scientific, Inc., Orem, UT, USA).

### 2.5. Preparation of Cell Extracts and Western Blot Analysis

Growth-arrested cells were treated with IL-1β at 37 °C for the indicated times. After treatment, the cells were washed with ice-cold PBS, scraped, and collected by centrifugation at 45,000× g for 1 h at 4 °C to yield the whole cell extract, as previously described [18]. Samples were analyzed by Western blot, transferred to NC (nitrocellulose) membrane, and then incubated overnight with an anti-phospho-c-Src, phospho-EGFR, phospho-Akt, phospho-ERK1/2, phospho-p38, phospho-JNK1/2, phospho-p65 NF-κB, or GAPDH antibody. Membranes were washed four times with TTBS for 5 min each and incubated with an anti-rabbit horseradish peroxidase antibody (1:2000) for 1 h. The immunoreactive bands were detected by ECL reagents and captured by a UVP BioSpectrum 500 Imaging System (Upland, CA, USA). The images were quantified and analyzed by an UN-SCAN-IT gel 6.1 software (Orem, UT, USA).

### 2.6. Measurement of Intracellular ROS Generation

The peroxide-sensitive fluorescent probe DCF-DA (2′,7′-dichlorofluorescein diacetate) was used to assess the production of intracellular ROS [25], with minor modifications. The bMECs were incubated with DCF-DA (5 μM) in RPMI-1640 at 37 °C for 45 min. The supernatant was removed and replaced with fresh RPMI-1640 medium before being incubated with IL-1β (10 ng/mL). Relative fluorescence intensity was detected at the indicated time by a fluorescent plate reader (Thermo, Appliskan, Vantaa, Finland) at an excitation (485 nm) and emission (530 nm).

### 2.7. Determination of NADPH Oxidase Activity by Chemiluminescence Assay

The NADPH oxidase activity was assayed by lucigenin chemiluminescence as described previously [25]. The cells were incubated with IL-1β for the indicated times and then gently scraped and centrifuged at 400 × g for 10 min at 4 °C. The cell pellet was resuspended in ice-cold RPMI 1640 medium (35 μL/well) and kept on ice. To a final 200 μL of pre-warmed (37 °C) RPMI 1640 medium containing either NADPH (1 μM) or lucigenin (20 μM), 5 μL of cell suspension (2 × 10^4^ cells) was added to initiate the reaction, followed by immediate measurement of chemiluminescence, using an Appliskan luminometer (ThermoW) in an out-of-coincidence mode. The blanks and controls were performed, and their chemiluminescence was detected. Neither NADPH nor NADH enhanced the background chemiluminescence of lucigenin alone (30–40 counts/min). Chemiluminescence was continuously measured for 12 min, and the activity of NADPH oxidase was presented as counts per million cells.

### 2.8. Immunofluorescence Staining

The cells were treated with 10 ng/mL IL-1β for the indicated times, washed twice with ice-cold PBS, fixed with 4% (*w*/*v*) paraformaldehyde in PBS for 30 min, and then permeabilized with 0.3% Triton X-100 in PBS for 15 min. The cells were stained by incubating with 10% normal goat serum in PBS for 30 min, followed by incubating with an anti-p65 NF-κB or ZO-1 antibody (1:200 dilution) for 1 h in PBS with 1% BSA, washing thrice with PBS, incubating with a FITC conjugated goat anti-rabbit antibody (1:200 dilution) in PBS with 1% BSA for 1 h, washing thrice with PBS, and mounting with aqueous mounting medium. The images were acquired under a fluorescence microscope (Axiovert 200 M, ZEISS, Göttingen Germany).

### 2.9. Promoter-Luciferase Reporter Gene Assay

The κB binding sites were cloned to the pGL4.32 (luc2P/NF-κB-RE/Hygro) vector containing the luciferase reporter system. The plasmids were prepared using QIAGEN plasmid DNA preparation kits. The cells were transfected with these constructs by using a Lipofectamine reagent according to the instructions of manufacture. The transfection efficiency (~60%) was assessed by transfection with enhanced GFP. After treatment with IL-1β, the cell lysates were collected in lysis buffer (25 mM Tris, pH 7.8, 2 mM EDTA, 1% Triton X-100, and 10% glycerol). After centrifugation, aliquots of the supernatants were detected for promoter activity, using a Dual-Luciferase^®^ Reporter Assay System (Promega, Madison, WI, USA). Firefly luciferase activities were standardized for renilla luciferase activity.

### 2.10. Statistical Analysis of Data

The data were estimated by using GraphPad Prism Program (GraphPad, San Diego, CA, USA). Quantitative data were analyzed by one-way ANOVA, followed by Tukey’s honestly significant difference tests between individual groups. Data were presented as mean ± SEM. A value of *p* < 0.05 was considered significant.

## 3. Results

### 3.1. Effects of Resveratrol, a Natural Polyphenol Compound, on IL-1β-Induced MMP-9 Expression in Brain Microvascular Endothelial Cells

Several cytokines, such as IL-1β, are crucial for various inflammatory disorders [27]. Previously, we have demonstrated that IL-1β induces the upregulation of MMP-9 in rat-brain astrocytes [18]. Here, we first investigated whether IL-1β can upregulate MMP-9 expression in brain microvascular endothelial cells (bMECs: bEnd.3), the cells were incubated with IL-1β (10 ng/mL) for the indicated time intervals, and then the conditioned media were collected and analyzed by gelatin zymography. As shown in Figure 1A, IL-1β (10 ng/mL) significantly induced MMP-9 expression in a time-dependent manner, and there was a significant increase between 16 and 24 h. The expressions of MMP-2 and a housekeeping protein, GAPDH, which was used as an internal control, were not changed. To further study whether the induction of MMP-9 expression by IL-1β resulted from the increase of MMP-9 mRNA expression, the RT-PCR analysis was performed. As shown in Figure 1B, IL-1β time-dependently induced the MMP-9 mRNA expression in these cells, whereas the β-actin mRNA, as an internal control, was not changed. A significant increase was achieved at 6 h and sustained until 24 h during the period of observation. These data revealed that the IL-1β-upregulated MMP-9 expression is mediated through the increasing mRNA level in bMECs. 

Furthermore, we determined the effects of resveratrol (a natural polyphenol compound) in IL-1β-induced MMP-9 expression; the cells were pretreated with resveratrol (Res, 1 μM, an optimal concentration according to our pre-tested results) for 1 h and then incubated with IL-1β for the indicated times. The gelatin zymography data showed that pretreatment with resveratrol significantly blocked IL-1β-induced MMP-9 expression (Figure 1C). Similarly, pretreatment with resveratrol also reduced IL-1β-induced MMP-9 mRNA expression by RT-PCR analysis (Figure 1D). These results indicated that resveratrol may behave as a neuroprotective potential by reducing IL-1β-induced MMP-9 expression in bMECs.

### 3.2. Role of Resveratrol in IL-1β-Induced MMP-9 Expression via Nox-Mediated ROS Production

Many studies have demonstrated that ROS contribute to MMP expression and inflammatory events in various cell types [16,28]. The NADPH oxidase (Nox) is a major source of ROS in many physiological and pathological processes [22,29]. A previous study demonstrated that Nox-derived ROS signaling cascades are involved in induction of MMP-9 by IL-1β in brain astrocytes [19]. Thus, to determine whether resveratrol inhibits IL-1β-induced upregulation of MMP-9 by reducing Nox-dependent ROS generation, the N-acetylcysteine (NAC, a ROS scavenger) and apocynin (a Nox activity inhibitor) were used. As shown in Figure 2A, NAC (10 mM) markedly attenuated IL-1β-induced MMP-9 expression in bMECs. Next, pretreatment with apocynin (Apo, 10 μM) also significantly inhibited IL-1β-induced MMP-9 expression in these cells (Figure 2A). The data suggested that Nox-derived ROS generation may be involved in IL-1β-induced MMP-9 expression in these cells. In brain astrocytes, IL-β significantly stimulates Nox-derived ROS generation within 30 min [18]. To confirm whether IL-1β stimulates Nox-derived ROS generation, the Nox activity and ROS production were detected in bMECs. As shown in Figure 2B, IL-1β stimulated a time-dependent increase of Nox activity with a significant activity at 10 min and sustained over 30 min. Pretreatment of cells with Apo (10 μM) attenuated IL-1β-stimulated Nox activity in these cells. Moreover, the cells were loaded with a ROS probe DCF-DA and then stimulated with IL-1β (10 ng/mL) for the indicated times. As expected, IL-1β stimulated ROS generation in a time-dependent manner and a maximal response at 10 min and maintained over 30 min (Figure 2C). Pretreatment of cells with NAC (10 mM) and Apo (1 μM) both markedly reduced IL-1β-stimulated ROS generation (Figure 2C). Additionally, to examine the effects of resveratrol on the IL-1β-stimulated Nox-derived ROS generation event, the cells were pretreated with resveratrol (Res, 1 μM) for 1 h and then incubated with IL-1β (10 ng/mL) for 30 min. As shown in Figure 2B,C, IL-1β-stimulated Nox activity and ROS generation both were markedly reduced by resveratrol, suggesting that resveratrol-decreased upregulation of MMP-9 by IL-1β may be through blocking Nox-derived ROS production in bMECs. Thus, we suggested that resveratrol may have an antioxidative effect in this event.

### 3.3. Resveratrol Attenuates IL-1β-Upregulated MMP-9 Expression through Blocking the Transactivation of EGFR in bMECs

The transactivation of receptor tyrosine kinases (RTKs), such as EGFR, by various stimuli has been indicated to contribute to several cellular functions, including inflammation [30]. Moreover, a previous report has pointed out that c-Src-dependent transactivation of RTKs such as PDGFR contribute to the upregulation of MMP-9 in IL-1β-challenged brain astrocytes [18]. Here, to explore whether c-Src-dependent transactivation of RTKs also participates in IL-1β-induced MMP-9 expression in bMECs, cells were pretreated with or without PP1 (1 μM, a c-Src inhibitor), AG1478 (30 μM, an EGFR inhibitor), or SH-5 (10 μM, an Akt inhibitor) for 1 h and then incubated with IL-1β (10 ng/mL) for the indicated time intervals. As shown in Figure 3A, pretreatment with PP1, AG1478, or SH-5 significantly blocked IL-1β-induced MMP-9 expression, suggesting that c-Src, EGFR, and Akt may be involved in this response. We further confirmed that IL-1β can stimulate the time-dependent phosphorylation of c-Src, EGFR, and Akt with a maximal response within 10 min by Western blot (Figure 3B). These results suggested that IL-1β-induced MMP-9 expression is mediated through c-Src, EGFR, and Akt in bMECs. Next, to determine whether IL-1β-stimulated c-Src-dependent transactivation of EGFR/Akt cascade and it is mediated through the ROS-dependent pathway, cells were pretreated with NAC, PP1, or AG1478 (AG) and then incubated with IL-1β (10 ng/mL) for 10 min. As shown in Figure 3A, pretreatment with PP1 (1 μM) significantly blocked IL-1β-stimulated phosphorylation of c-Src, ERGF, and Akt. Pretreatment with AG1478 (AG, 30 μM) significantly inhibited IL-1β-stimulated phosphorylation of ERGF and Akt, but not c-Src. These data suggested that IL-1β stimulated transactivation of EGFR/Akt cascade via c-Src-dependent manner in bMECs. Moreover, we further found that pretreatment with NAC (10 mM) significantly attenuated IL-1β-stimulated phosphorylation of c-Src, ERGF, and Akt (Figure 3B), suggesting that IL-1β stimulated c-Src-dependent transactivation of EGFR/Akt cascade through ROS-mediated pathway in these cells. Subsequently, we further evaluated the effect of resveratrol in IL-1β-stimulated c-Src-dependent transactivation of EGFR/Akt pathway. Similarly, pretreatment with resveratrol (Res, 1 μM) markedly decreased IL-1β-stimulated phosphorylation of c-Src, EGFR, and Akt (Figure 3B). These results demonstrated that resveratrol inhibited IL-1β-induced MMP-9 expression is mediated through blocking ROS-mediated activation of c-Src-dependent transactivation of EGFR/Akt cascade in bMECs.

### 3.4. Effects of Resveratrol on IL-1β-Induced MMP-9 Expression via the ROS-Mediated c-Src-Dependent EGFR Linking to MAPKs Cascade

Activation of MAPKs could regulate several functions of brain cells [31]. Moreover, many reports have shown that MAPKs, including ERK1/2, p38, and JNK1/2, are critical for the IL-1β-upregulated MMP-9 in brain astrocytes [18]. Thus, to determine whether MAPKs participate in IL-1β-induced MMP-9 expression, the bMECs were pretreated with or without U0126 (1 μM), SB202190 (SB, 30 μM), or SP600125 (SP, 10 μM) for 1 h and then incubated with IL-1β (10 ng/mL) for the indicated time intervals. As shown in Figure 4A, pretreatment with U0126, SB202190, or SP600125 attenuated IL-1β-induced MMP-9 expression, suggesting that MAPKs (i.e., ERK1/2, p38, and JNK1/2) may be involved in IL-1β-induced MMP-9 expression in bMECs. We further demonstrated that IL-1β stimulated the time-dependent phosphorylation of MAPKs (i.e., ERK1/2, p38, and JNK1/2) with a maximal response at 5 min and maintained over 15 min (Figure 4B), and this was attenuated by pretreatment with respective specific inhibitor of MAPKs by Western blot analysis. These results suggested that IL-1β-induced MMP-9 expression is mediated via MAPKs (i.e., ERK1/2, p38, and JNK1/2) pathways in bMECs. Next, to determine whether IL-1β-stimulated phosphorylation of MAPKs is mediated through the ROS-mediated c-Src-dependent EGFR cascade, cells were pretreated with NAC (10 mM), PP1 (1 μM), or AG1478 (AG, 30 μM) and then incubated with IL-1β (10 ng/mL) for 15 min. As shown in Figure 4C, pretreatment with NAC, PP1, or AG1478 significantly inhibited IL-1β-stimulated phosphorylation of MAPKs (i.e., ERK1/2, p38, and JNK1/2), suggesting that IL-1β stimulated the phosphorylation of MAPKs (i.e., ERK1/2, p38, and JNK1/2) via the ROS-mediated activation of c-Src/EGFR pathway in these cells. Furthermore, we evaluated the effect of resveratrol in the IL-1β-stimulated phosphorylation of MAPKs (i.e., ERK1/2, p38, and JNK1/2). The results showed that the pretreatment with resveratrol (Res, 1 μM) markedly reduced IL-1β-stimulated phosphorylation of MAPKs (i.e., ERK1/2, p38, and JNK1/2) (Figure 4C), suggesting that resveratrol inhibited IL-1β-induced MMP-9 expression by blocking the activation of MAPKs, including ERK1/2, p38, and JNK1/2 in these cells. These results demonstrated that resveratrol inhibited IL-1β-induced MMP-9 expression, which is mediated via attenuating ROS/c-Src/EGFR-mediated activation of MAPKs in bMECs.

### 3.5. Resveratrol Reduces IL-1β-Induced MMP-9 Expression by Blocking Activation of the Transcription Factor NF-κB in bMECs

The NF-κB-dependent pathways are involved in MMP-9 expression in several cell types [16]. Here, we determine whether IL-1β-induced MMP-9 expression is mediated via activation of NF-κB in bMECs. As shown in Figure 5A, pretreatment with Bay11-7082 (Bay, 1 μM), a NF-κB inhibitor, significantly inhibited IL-1β-induced MMP-9 expression, suggesting that NF-κB may participate in IL-1β-induced MMP-9 expression in bMECs. Furthermore, the results of the Western blotting analysis showed that IL-1β induced the time-dependent phosphorylation of p65 (a subunit of NF-κB), a significant increase within 5~15 min (Figure 5B). Pretreatment with Bay (1 μM), U0126 (U0, 1 μM), SB202190 (SB, 30 μM), and SP600125 (SP, 10 μM) attenuated IL-1β-stimulated p65 NF-κB phosphorylation (Figure 5C). These results suggested that IL-1β induced MMP-9 expression via MAPKs-mediated (i.e., ERK1/2, p38, and JNK1/2) activation of NF-κB cascade in bMECs. Next, we explored the effects of resveratrol on IL-1β-stimulated p65 NF-κB phosphorylation; as shown in Figure 5C, resveratrol (Res, 1 μM) significantly reduced IL-1β-stimulated p65 NF-κB phosphorylation. Additionally, we also confirmed that IL-1β stimulated translocation of p65 NF-κB into the nucleus in a time-dependent manner by immunofluorescence staining (Figure 5D). Pretreatment with resveratrol (Res, 1 μM) attenuated IL-1β-stimulated p65 NF-κB translocation in bMECs (Figure 5E). Moreover, previous reports indicated that NF-κB binding sites are contained in the MMP-9 promoter region [18]. Therefore, we further examined whether resveratrol affects IL-1β-stimulated transcriptional activity of NF-κB; a promoter reporter construct (pGL4.32-contains κB binding sites) was used. The data showed that IL-1β can induce NF-κB transcriptional activity at 6 h, and it was significantly attenuated by pretreatment with Bay11-7082 (Bay, 1 μM) or resveratrol (Res, 1 μM) in these cells (Figure 5F). These results indicated that IL-1β-induced MMP-9 expression is mediated via MAPKs-mediated activation of NF-κB in bMECs. Moreover, resveratrol may play a blocker in IL-1β-induced MMP-9 expression through inhibiting NF-κB-mediated events, including p65 NF-κB phosphorylation, translocation, and transcriptional activity, in these cells.

### 3.6. Resveratrol Mitigates the MMP-9-Mediated Disruption of Tight Junction Protein ZO-1 Arranged Integrity in IL-1β-Challenged Brain Microvascular Endothelial Cells

The MMP-9 can be upregulated in several brain injuries and participates in the pathogenesis of various CNS disorders, including brain inflammation and BBB damage [17]. Hence, we next investigated the effects of resveratrol on IL-1β-induced MMP-9-mediated endothelial cell functions such as the disruption of tight junction protein ZO-1 arranged integrity. First, the images of cellular ZO-1 staining were performed and taken 24 h after IL-1β (10 ng/mL) treatment. We found that ZO-1 was completely arranged along the cell membrane, and this arranged integrity was disrupted after treatment of IL-1β (Figure 6A). To confirm the role of MMP-9 in this event, an MMP-9 inhibitor was used. As shown in Figure 6A, pretreatment with the MMP-9 inhibitor (9i, 1 μM) significantly inhibited IL-1β-induced disruption of ZO-1 arranged integrity. Moreover, to further check the expression of MMP-9 in this situation, the conditioned media of the immunofluorescence staining experiment were collected and analyzed by gelatin zymography. As expected, pretreatment with MMP-9 inhibitor (MMP9i, 1 μM) markedly attenuated IL-1β-induced MMP-9 expression and activity (Figure 6B). These data demonstrated that the IL-1β induces MMP-9-mediated disruption of ZO-1 arranged integrity in bMECs. Next, to evaluate the effect of resveratrol on this response, the cells were pretreated with resveratrol (Res, 1 μM) and then incubated with IL-1β for 24 h. The data showed that IL-1β-induced MMP-9-mediated disruption of ZO-1 arranged integrity was inhibited by pretreatment of resveratrol (Figure 6C). The results suggested that resveratrol repressed IL-1β-induced disruption of ZO-1 arranged integrity via reducing MMP-9 expression in brain microvascular endothelial cells.

## 4. Discussion

MMPs participates in a wide range of biological activities in diverse tissues, including several CNS disorders, such as stroke, malignant glioma, and Alzheimer’s disease [14,15,32]. MMP-9 particularly plays an important role in tissue remodeling and in the pathogenesis of brain diseases [14,15,32]. Moreover, MMP-9 has been shown to be a risk factor in the brain–blood barrier (BBB) integrity [33]. The destruction of the BBB is a critical feature associated with neuroinflammatory conditions in the CNS disorders. The cerebrovascular cells, such as brain microvascular endothelial cells, are critical for maintaining the brain microenvironment and functions, such as the BBB functional integrity [1]. Attenuation of MMP activity by pharmacological inhibitors or gene-knockout strategies protects the brain from BBB destruction, cell death, and advanced neuroinflammation [33,34]. Moreover, several proinflammatory cytokines such as IL-1β can induce various inflammatory factors, including MMP-9 following brain injury [27,35]. We have demonstrated that IL-1β-induced upregulation of MMP-9 in brain astrocytes, which caused astrocytic functional changes, such as cell motility [19]. These studies suggested that the upregulation of MMP-9 by proinflammatory mediators in brain cells may be a crucial effect upon brain injury and BBB damage that may provide a therapeutic strategy to brain inflammation and neurodegenerative diseases. Pharmacological and knockout-mouse approaches indicate that targeting MMP-9 and its upstream signaling pathways should yield useful therapeutic targets for brain injury, tumor, and CNS inflammatory disorders. Herein, we first found that IL-1β can induce MMP-9 expression and further demonstrated the signaling mechanisms and effects in brain microvascular endothelial cells (bMECs). Next, we also evaluated the natural product resveratrol to determine its antioxidative and neuroprotective effects on IL-1β-induced MMP-9-related events and its mechanism. The results demonstrate that, in bMECs, resveratrol inhibited IL-1β-induced the MMP-9-mediated disruption of ZO-1 arranged integrity by reducing the ROS-signal-activated c-Src-dependent EGFR/MAPKs (ERK1/2, p38, and JNK1/2) cascade, leading to the activation of the NF-κB pathway.

Here, we found that IL-1β can upregulate MMP-9 gene and protein expression in bMECs (bEnd.3) that was inhibited by a natural polyphenolic compound, resveratrol (Figure 1). It is consistent with the report that showed that resveratrol suppresses IL-1β-induced inflammatory signaling, including MMP-9 in human articular chondrocytes [36]. However, our result is the first finding that resveratrol can reduce MMP-9 upregulation by IL-1β in the bMECs (bEnd.3). Next, we investigated the signaling mechanisms of IL-1β-induced MMP-9 expression in bMECs. Among them, redox imbalance is a cause for various pathologies of degenerative diseases [16]. ROS plays a pivotal role in the normal physiological functions and the inflammatory disorders by concentration-dependent manner [20,37]. In the brain, ROS also exert the control of vascular tone, which is tightly modulated by metabolic activity within neurons. Furthermore, ROS production by diverse stimuli can affect several inflammatory genes’ expression in the pathogenesis of brain disorders [16,38]. Recently, the cell damage in neurodegenerative disorders, such as Alzheimer’s disease (AD) and Parkinson’s disease (PD), is attributed to oxidative stress in brain inflammatory disorders [23,39]. Previously, we have indicated that, in both in vitro and in vivo studies, the ROS-related signals participate in the upregulation of MMP-9 by some stimuli, including IL-1β in brain astrocytes [16,19]. Similarly, in bMECs (bEnd.3), we are the first group to demonstrate that IL-1β-induced MMP-9 expression is mediated through Nox/ROS-dependent pathway (Figure 2). Moreover, we further demonstrated that resveratrol may possess an antioxidative activity by reducing IL-1β-stimulated Nox-derived ROS signal in bMECs (Figure 2). This is also the first study to establish that resveratrol inhibits IL-1β-induced ROS-signal-mediated MMP-9 expression in bMECs. The result is consistent with a previous report that indicated that resveratrol can inhibit ROS generation and MMP-9 upregulation in manganese (Mn)-induced rat astrocytes [40], in lipopolysaccharide (LPS)-stimulated human gingival fibroblasts [41], and in IL-1β-stimulated rat chondrocytes [42]. In contrast with the previous study, resveratrol enhanced the temozolomide-mediated antitumor effects in glioblastoma by ROS-dependent AMPK-TSC-mTOR signaling pathway [43].

Previous reports indicated that an alternative signaling pathway is critical for a variety of biological responses. The cross-communication between heterologous signaling systems and protein tyrosine kinases (PTKs), such as c-Src, epidermal growth factor receptor (EGFR), or platelet-derived growth factor receptor (PDGFR), is involved in various cell types, which can activate downstream signaling pathways such as the PI3K/Akt cascade [44,45]. Previous studies have demonstrated that the transactivation of EGFR or PDGFR is involved in proinflammatory cytokines (e.g., IL-1β) and induced several inflammatory mediators, such as MMPs in rat astrocytes [19] and human keratinocytes [46]. Moreover, the transactivation of EGFR has also been indicated to participate in ET-1-induced COX-2 expression in brain microvascular endothelial cells [27]. Similarly, we found that IL-1β-induced MMP-9 expression was blocked by an EGFR antagonist AG1478 (Figure 3), demonstrating that, in bMECs, c-Src-dependent transactivation of EGFR and PI3K/Akt cascade is involved in the response. The result is also consistent with the recent report that demonstrated IL-1β-induced tissue factor expression via EGFR-dependent and -independent mechanisms in lung cancer A549 cells [47]. Moreover, several studies have indicated that the transactivation of EGFR can occur through Nox/ROS-dependent manner [45]. Here, we also demonstrated that IL-1β stimulates c-Src-dependent transactivation of EGFR through an ROS-mediated manner in bMECs (Figure 3). Consistent with these results, in VSMCs and keratinocytes, Nox-derived ROS production leads to the transactivation of EGFR [48,49]. Moreover, pretreatment of resveratrol can attenuate IL-1β-stimulated phosphorylation of c-Src, EGFR, and Akt in bMECs (Figure 3), suggesting that resveratrol inhibits IL-1β-induced MMP-9 expression via reducing ROS-mediated c-Src-dependent transactivation of EGFR/Akt cascade in bMECs.

MAPKs are the second messengers to regulate signal transduction and gene expression, which trigger an important inflammatory event via MAPK-mediated cascades in various cell types [50,51]. The activity of MAPKs is abnormally regulated in several CNS inflammation and injury models [50,51]. Several studies also indicated that IL-1β induced MMP-9 expression in various cells via the MAPKs-dependent pathways. Moreover, we have demonstrated that activation of MAPKs, including ERK1/2, p38, and JNK1/2, is required for the upregulation of MMP-9 by IL-1β in brain astrocytes [18,19]. Here, our results showed that the MAPKs (i.e., ERK1/2, p38, and JNK1/2) also participated in IL-1β-induced MMP-9 expression in bMECs (Figure 4) that was activated via ROS-mediated c-Src-dependent transactivation of the EGFR pathway. Moreover, resveratrol attenuated IL-1β-stimulated phosphorylation of ERK1/2, p38, and JNK1/2 in these cells (Figure 4). The results are consistent with the report that indicated that IL-1β induces MMP-9 expression via ROS-dependent MAPKs in rat-brain astrocytes [19]. The results also demonstrated that resveratrol can reduce IL-1β-induced MMP-9 expression by blocking the ROS-mediated activation of c-Src/EGFR/MAPKs cascades in bMECs. These findings suggested that ROS-derived MAPKs’ activation may be a potent molecular target for novel therapy in the CNS disorders [52]. Moreover, the resveratrol data are consistent with the report that showed that resveratrol inhibits LPS-induced inflammation by suppressing the signaling cascade of TLR4–MAPKs [53].

The progressive increase of redox stress during injuries not only causes cellular oxidative damage, but it also regulates the pattern of gene expression via diverse transcription factors. Among these transcription factors, NF-κB plays a critical role in the regulation of different gene expressions, such as MMP-9, during the inflammation, cell proliferation, and apoptosis associated with physiological and pathological events [16,54]. In brain inflammation, several stimuli can induce the expression of diverse inflammatory mediators, including MMP-9 by ROS-mediated activation of the NF-κB cascade in brain cells [16]. Our results have indicated that NF-κB is involved in the regulation of diverse genes such as MMP-9 by IL-1β via an ROS-dependent manner in brain astrocytes [19]. These results implicate that NF-κB plays a critical role in MMP-9 induction and inflammatory genes expression in pathological events such as CNS inflammation [16,55]. Therefore, we investigated the effects of resveratrol on IL-1β-stimulated activation of NF-κB in bMECs. Here we found that NF-κB participates in IL-1β-induced MMP-9 expression (Figure 5). Moreover, the Nox/ROS, c-Src-dependent transactivation of EGFR, and MAPKs (i.e., ERK1/2, p38, and JNK1/2) cascades were involved in IL-1β-stimulated NF-κB activation. IL-1β-stimulated NF-κB activation, including p65/NF-κB phosphorylation, nuclear translocation, and transcriptional activity, but it was significantly inhibited by resveratrol (Figure 5). These results indicated that resveratrol may reduce the upregulation of MMP-9 by IL-1β through inhibiting the activation of NF-κB in bMECs. This result is consistent with the result that showed that resveratrol alleviates early brain injury after subarachnoid hemorrhage via blocking the NF-κB-dependent inflammatory/MMP-9 pathway [55]. Finally, our data showed that IL-1β induces MMP-9 expression via ROS-mediated c-Src/EGFR and MAPKs signals, linking to activation of NF-κB, which results in the brain microvascular endothelial cells (bMECs) ZO-1 arranged integrity disruption (representing BBB destruction) (Figure 6). In the study, we found that resveratrol can alleviate this MMP-9-mediated event by inhibiting a series of signaling pathways in the bMECs mentioned in the text. 

In conclusion, based on the results from the literature and our data, Figure 7 illustrates a model for the inhibitory effect of resveratrol on IL-1β-induced MMP-9-mediated events such as the disruption of ZO-1 arranged integrity in bMECs. In the study, the results showed that resveratrol markedly attenuates the activation of related signaling factors in IL-1β-induced MMP-9 expression, including Nox/ROS, c-Src-dependent transactivation of EGFR, MAPKs (i.e., ERK1/2, p38, and JNK1/2), and NF-κB. These findings that the natural polyphenolic compound resveratrol reduced IL-1β-induced ROS-mediated c-Src/EGFR/MAPKs (i.e., ERK1/2, p38, and JNK1/2), NF-κB, and MMP-9-dependent disruption of ZO-1 arranged integrity (tight junction disruption) in brain microvascular endothelial cells indicated that resveratrol may have antioxidative, anti-inflammatory, and neuroprotective properties in brain inflammatory disorders.

## Figures and Tables

**Figure 1 biomedicines-10-01270-f001:**
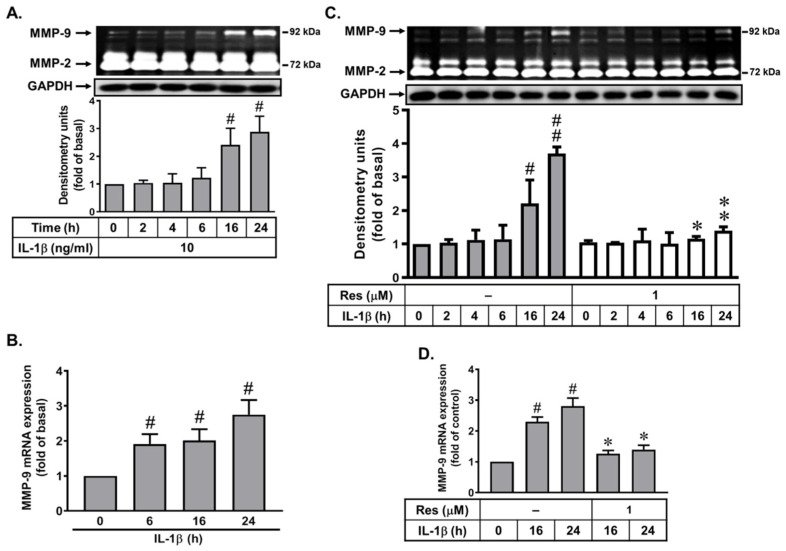
Effects of resveratrol on IL-1β-induced MMP-9 expression in bEnd.3 cells, including protein and mRNA level. (**A**,**B**) Time dependence of IL-1β increase of MMP-9 expression. The bEnd.3 cells were treated with 10 ng/mL IL-1β for the indicated time intervals. (**C**,**D**) Cells were pretreated with resveratrol (Res, 1 μM) for 1 h and then incubated with IL-1β (10 ng/mL) for the indicated times. After treatment, the conditioned media, cell lysates, and total RNA were collected and analyzed by gelatin zymography (MMP2/9), Western blot (GAPDH, as an internal control), and RT-PCR (*MMP-9* and *β-actin*), as described in Methods. The intensity of zymographic (**A**,**C**) and PCR product (**B**,**D**) bands were quantitated by scanning densitometry and expressed as fold of untreated control. Quantitative data were analyzed by one-way ANOVA, as described in Methods. Data are expressed as the mean ± SEM (*n* = 3); # *p* < 0.05; ## *p* < 0.01, as compared with the untreated control; * *p* < 0.05; ** *p* < 0.01, as compared cells stimulated with IL-1β only (**C,D**). The image represents one of three individual experiments.

**Figure 2 biomedicines-10-01270-f002:**
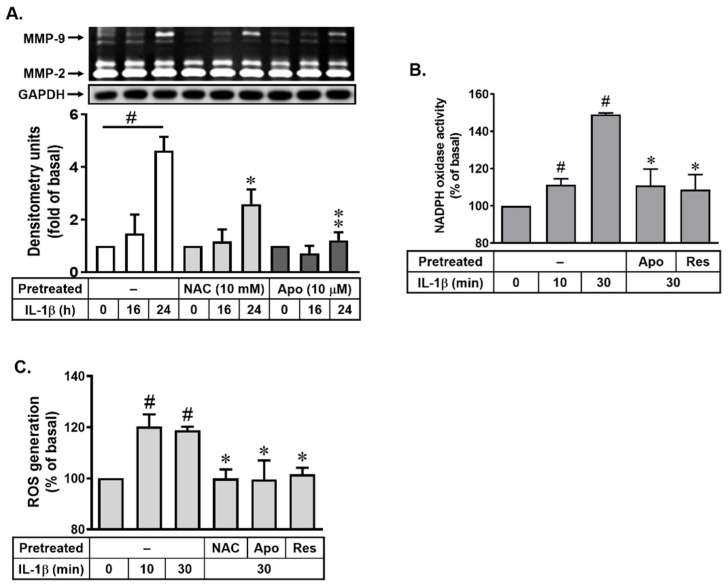
Roles of resveratrol in IL-1β-induced MMP-9 expression via Nox-mediated ROS generation. (**A**) The Nox/ROS system inhibitors blocked IL-1β-induced MMP-9 expression; cells were pretreated with NAC (10 mM) or apocynin (Apo, 10 μM) and then incubated with IL-1β (10 ng/mL) for 0, 16, and 24 h. (**B**) IL-1β stimulates Nox activity increase. Cells were pretreated without or with Apo (10 μM) or Res (1 μM) for 1 h before exposure to IL-1β (10 ng/mL) for the indicated times. (**C**) IL-1β stimulates Nox-mediated ROS generation. Cells were pretreated without or with NAC (10 mM), Apo (10 μM), or Res (1 μM) for 1 h before exposure to IL-1β (10 ng/mL) for the indicated times. The conditioned media were collected and analyzed as described in Figure 1A. The Nox activity (**B**) and ROS generation (**C**) were analyzed, as described in Methods. Quantitative data were analyzed by one-way ANOVA, as described in Methods. Data are expressed as the mean ± SEM (*n* = 3); ^#^
*p* < 0.01, as compared with the untreated control; * *p* < 0.05; ** *p* < 0.01, as compared with the cells stimulated with IL-1β only. The image represents one of three individual experiments.

**Figure 3 biomedicines-10-01270-f003:**
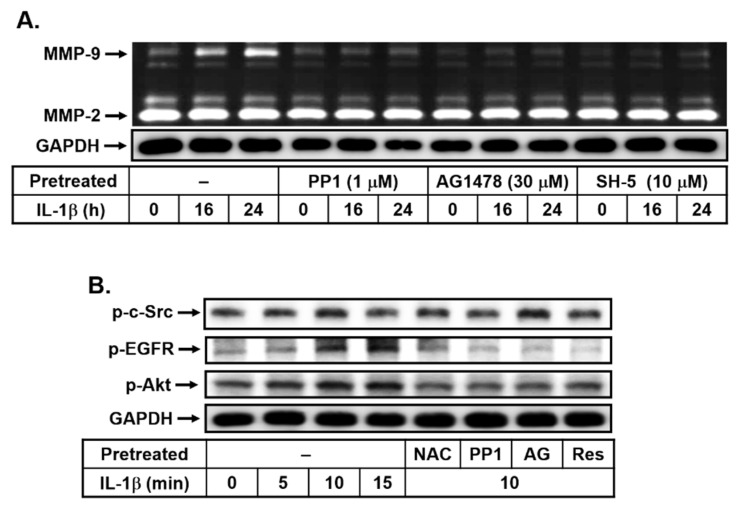
Resveratrol inhibits IL-1β-induced MMP-9 expression via blocking Nox/ROS-mediated c-Src-dependent transactivation of EGFR in bEnd.3 cells. (**A**) The c-Src-dependent transactivation of EGFR was involved in IL-1β-induced MMP-9 expression; cells were pretreated with PP1 (1 μM), AG1478 (30 μM), and SH-5 (10 μM) and then stimulated with IL-1β (10 ng/mL) for 0, 16, and 24 h. (**B**) IL-1β stimulates a c-Src-dependent transactivation of EGFR/Akt pathway. Cells were pretreated without or with NAC (10 mM), PP1 (1 μM), AG1478 (AG, 30 μM), and Res (1 μM) for 1 h before exposure to IL-1β (10 ng/mL) for the indicated times. The conditioned media were collected and analyzed as described in Figure 1A. The cell lysates were analyzed for phosphorylation of c-Src (p-c-Src), EGFR (p-EGFR), Akt (p-Akt), and GAPDH by Western blot, as described in Methods. The image represents one of three individual experiments.

**Figure 4 biomedicines-10-01270-f004:**
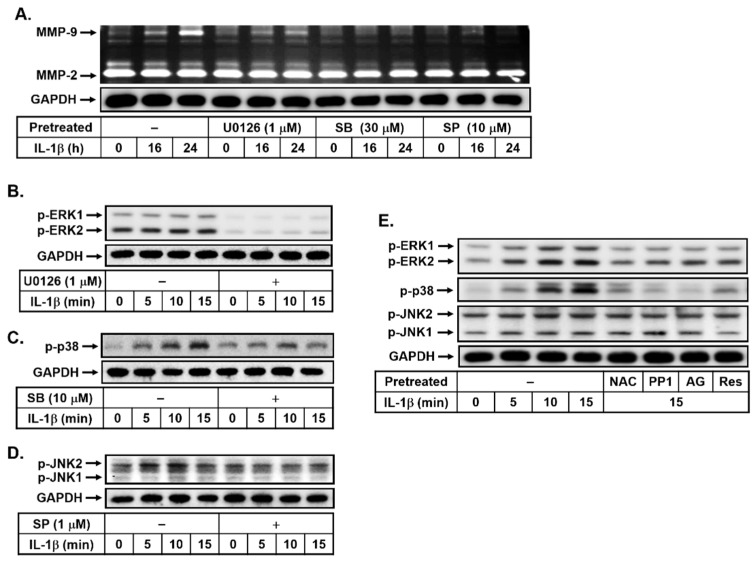
Resveratrol attenuates IL-1β-induced MMP-9 expression through repressing MAPK-dependent manner, including ERK1/2, p38, and JNK1/2. (**A**) Cells were pretreated with U0126 (1 μM), SB202190 (30 μM), and SP600125 (10 μM) for 1 h and then incubated with IL-1β (10 ng/mL) for the indicated time intervals. (**B**–**D**) Cells were pretreated with (**B**) 1 μM U0126, (**C**) 30 μM SB202190 (SB), and (**D**) 10 μM SP600125 (SP) for 1 h and then incubated with IL-1β (10 ng/mL) for the indicated time intervals. (**E**) Cells were pretreated with NAC (10 mM), PP1 (1 μM), AG (30 μM), and Res (1 μM) for 1 h before exposure to IL-1β (10 ng/mL) for the indicated times. The conditioned media were collected and analyzed as described in Figure 1A. The cell lysates were analyzed for the phosphorylation of (**B**,**E**) ERK1/2 (p-ERK1/2), (**C**,**E**) p38 MAPK (p-p38), (**D**,**E**) JNK1/2 (p-JNK1/2), and (**A**–**E**) GAPDH by Western blot, as described in Methods. The image represents one of three individual experiments.

**Figure 5 biomedicines-10-01270-f005:**
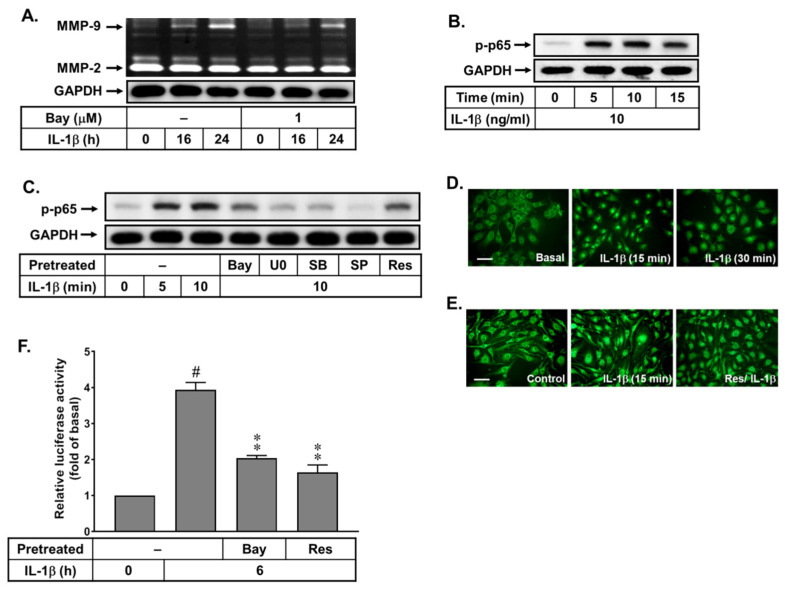
Effects of resveratrol on IL-1β-induced MMP-9 expression via activation of NF-κB in bEnd.3 cells. (**A**) Cells were pretreated with Bay11-7082 (Bay, 1 μM) and then incubated with IL-1β (10 ng/mL) for the indicated times. The conditioned media were collected and analyzed as described in Figure 1. (**B**) Time dependence of IL-1β increased phosphorylation of NF-κB; cells were treated with 10 ng/mL IL-1β for the indicated time intervals. (**C**) Cells were pretreated with Bay (1 μM), U0126 (U0, 1 μM), SB (30 μM), SP (10 μM), and Res (1 μM) for 1 h and then stimulation with IL-1β (10 ng/mL) for the indicated time intervals. The cell lysates were collected and analyzed phosphorylation of p65 NF-κB (p-p65) (**B**,**C**) and GAPDH (**A**–**C**) by Western blot, as described in Methods. (**D**) IL-1β stimulates nuclear translocation of NF-κB; cells were stimulated with 10 ng/mL IL-1β for the indicated time intervals. (**E**) Cells were pretreated with Res (1 μM) for 1 h and then were stimulated with IL-1β for 15 min. Cells were fixed and labeled with anti-p65 NF-κB antibody and a fluorescein isothiocyanate (FITC)-conjugated secondary antibody. Individual cells were imaged ((**D**,**E**), scale bar = 50 μm), as described in Methods. (**F**) Quantitative data were analyzed by one-way ANOVA, as described in Methods. Data are expressed as the mean ± SEM (*n* = 3); ^#^
*p* < 0.01, as compared with the untreated control; ** *p* < 0.01, as compared with the cells stimulated with IL-1β only. The image represents one of three individual experiments.

**Figure 6 biomedicines-10-01270-f006:**
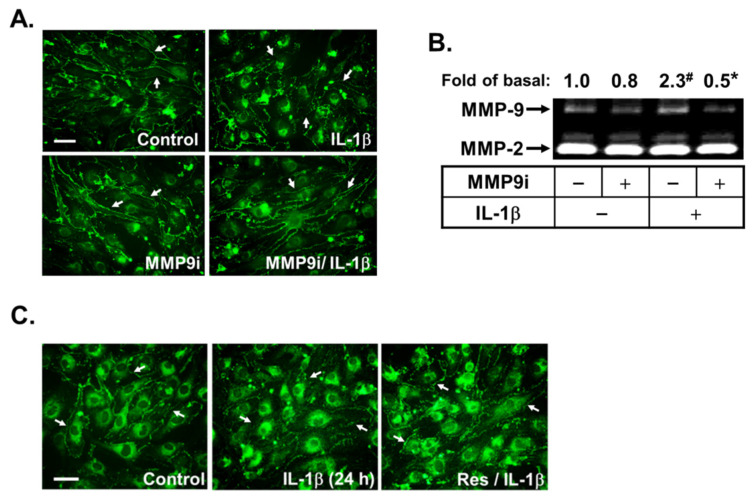
Effect of resveratrol on MMP-9-mediated destruction of tight junction protein ZO-1 arranged integrity in response to IL-1β treatment. (**A**) Cells grown on coverslips were pretreated with MMP-9 inhibitor (MMP9i, 1 μM) for 1 h and then exposed to IL-1β (10 ng/mL) for 24 h. (**B**) After treatment, the conditioned media were collected and assayed as described in Figure 1. Quantitative data were analyzed by one-way ANOVA, as described in Methods. Data are expressed as the mean ± SEM (*n* = 3); ^#^
*p* < 0.01, as compared with the untreated control; * *p* < 0.05, as compared with the respective values of cells stimulated with IL-1β only. (**C**) Cells grown on coverslips were pretreated with resveratrol (Res, 1 μM) for 1 h and then exposed to IL-1β (10 ng/mL) for 24 h. Cells were fixed and stained with anti-ZO-1 antibody and a fluorescein isothiocyanate (FITC)-conjugated secondary antibody (**A**,**C**). The image represents one of three individual experiments ((**A**,**C**), scale bar = 20 μm). The white arrows point to the location of ZO-1 around the cell membrane.

**Figure 7 biomedicines-10-01270-f007:**
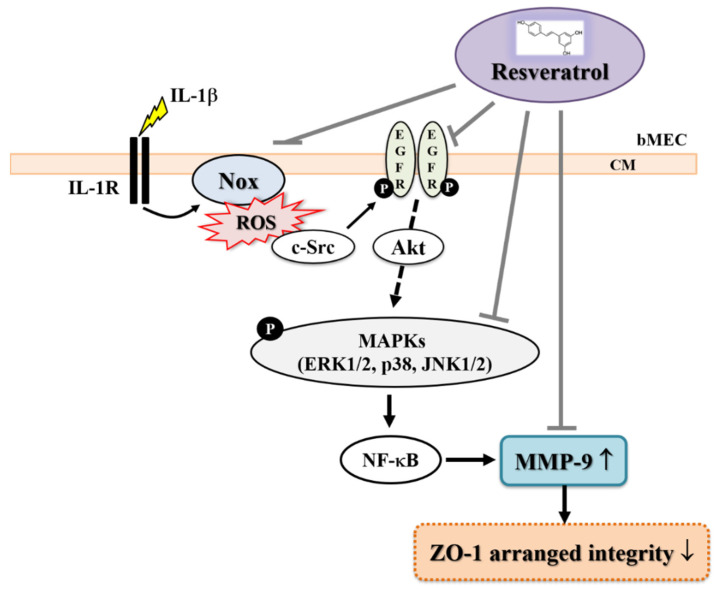
Schematic presentation of the effects of resveratrol on the IL-1β-induced upregulation of MMP-9 and disruption of ZO-1 arranged integrity in bEnd.3 cells. In brain microvascular endothelial cells (bEnd.3 cells), IL-1β induces MAPKs (i.e., ERK1/2, p38, and JNK1/2) activation through Nox-derived ROS signal and c-Src-mediated transactivation of EGFR pathways, resulting in NF-κB-dependent MMP-9 expression. The raised MMP-9 (MMP-9↑) causes disruption of ZO-1 arranged integrity (ZO-1 arranged integrity↓). Resveratrol blocks the IL-1β-induced MMP-9-mediated events (disruption of ZO-1 arranged integrity) via inhibiting activation of Nox/ROS, c-Src-dependent transactivation of EGFR, MAPKs (i.e., ERK1/2, p38, and JNK1/2), and NF-κB signaling pathways in these cells.

## Data Availability

The data presented in this study are available in this article.

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
