# Peer review of "Brain Protective Effect of Resveratrol via Ameliorating Interleukin-1β-Induced MMP-9-Mediated Disruption of ZO-1 Arranged Integrity"

_biomedicines, 2022, doi:10.3390/biomedicines10061270_

Round 1

Reviewer 1 Report

In this study the Authors provide interesting insights on the protective effect of resveratrol on brain microvascular endothelial cells (bMECs). They found that IL-1beta can upregulate MMP-9 gene and protein expression in bMECs, and that this effect was inhibited by resveratrol. Moreover, by using several pharmacological inhibitors, the Authors unveil a novel molecular mechanism by which IL-1beta induced MMP-9 expression via ROS-mediated c-Src-dependent transactivation of EGFR, and then activation of ERK1/2, p38 MAPK, and JNK1/2, 39 NF-kappa B signaling pathway. Finally, the Authors demonstrated that resveratrol can alleviate this MMP-9-mediated event.

The writing is fine making the manuscript very easy to follow. I have only some specific comments. 

- Regarding figure legends please specify the used statistical test for each analysis.

- For each reagent/antibody/chemical used the Authors must provide the producer and Country.

- Immunofluorescence images: please add scale bar.

- Figure 6: please specify the use of the arrows in the figure legend.

- Figure 6C, as per Fig. 6A, please add arrows.

- Have the Authors used other dyes (e.g. dihydroethidium) for intracellular ROS detection?

- Lines 558-560: the sentence is not clear. Please revise it.

- Line 671: please correct.

Reviewer 2 Report

Thank you for the opportunity to review this manuscript entitled ‘Brain Protective Effect of Resveratrol via Ameliorating Interleukin-1beta-Induced MMP-9-Mediated Disruption of ZO-1 Arranged Integrity’ by Tsai and co-authors.

In the present study, authors performed to evaluate the effects and signaling mechanisms of resveratrol on IL-1beta-induced MMP-9 expression in brain microvascular endothelial cells.

The pathological changes in the integrity and function of brain microvessels can trigger the development or dramatic progression neurodegenerative diseases, including Alzheimer's disease that is becoming an urgent problem in connection with the increase in the proportion of the elderly population in developed countries.

The manuscript is well written and logical and presents data appropriate for publication in Biomedicines journal. However, there are some key points that require clarification.

  1. Why didn't you use phosphatase inhibitors to obtain protein homogenates? This is important when working with phosphorylated forms of proteins.
  2. Add the histogram of gelatin zymography analysis results for MMP-9 in figure 1C.
  3. Why didn't you include data the mRNA MMP-9 expression in time interval of 24 hours in Fig.1D? Add these results please.
  4. Fig.1A, C does not include molecular weight data of MMP-2 and MMP-9.
  5. In the pretreatment (NAC and Apo) experiment, you used different time intervals to determine the MMP-9 and MMP-2 expressions, NADPH oxidase activity and ROS generation. Explain your choice and add this to the manuscript. This remark applies to all subsequent experiments.
  6. There is no description in the Fig.6A caption what the white arrows point to. It's important to add.

Reviewer 3 Report

This study expand our knowledge on intracellular mechanism of protective effects of resveratrol in oxidative-stress-related brain damages. Specifically, the authors in well-designed series of in vitro experiments obtained a convincing evidence that resveratrol inhibits the IL-beta-induced MMP-9-mediated disruption of zonula occludens-1 (ZO-1) arranged integrity in brain microvascular endothelial cells (bMECs). Furthermore, they found that resveratrol blocks the IL-1beta-induced disruption of ZO-1 arranged integrity via inhibiting activation of Nox/ROS, c-Src-dependent transactivation of EGFR, MAPKs (i.e., ERK1/2, p38, and JNK1/2), and NF-kB signaling pathways. All biochemical tests are sound and were properly performed. Some results support previous findings, but majority of them are  quite original, and they  have been clearly presented and described. The graphical abstract is also well constructed. Discussion and conclusions are supported by the obtained data.

Other remarks:

  1. The potential role of iNOS and nitric oxide in mechanism of resveratrol effects on tight junction proteins ZO-1 should be mentioned in the Introduction or Discussion. As reported by Wang et al.,(J Neurophysiol. 2016 Nov 1;116(5):2173-2179) resveratrol suppressed the EAE-induced overexpression of proinflammatory transcripts iNOS and IL-1β and upregulated the expression of anti-inflammatory transcripts arginase 1 and IL-10 cytokine in the brain.
  2. The rationale for using resveratrol only in one concentration (1uM) should be given. How was this compound resolved?
